# CD5-Negative, CD10-Negative Low-Grade B-Cell Lymphoproliferative Disorders of the Spleen

John J. Schmieg [1], Jeannie M. Muir [1], Nadine S. Aguilera [2] and Aaron Auerbach [1,*]

1    The Joint Pathology Center, Silver Spring, MD 20910, USA; john.j.schmieg.civ@mail.mil (J.J.S.); jeannie.m.muir.mil@mail.mil (J.M.M.)
2    Department of Pathology, University of Virginia Health System, Charlottesville, VA 22904, USA; na2d@virginia.edu
*    Correspondence: aaron.auerbach.civ@mail.mil; Tel.: +1-301-295-5636

**Abstract:** CD5-negative, CD10-negative low-grade B-cell lymphoproliferative disorders (CD5-CD10-LPD) of the spleen comprise a fascinating group of indolent, neoplastic, mature B-cell proliferations that are essential to accurately identify but can be difficult to diagnose. They comprise the majority of B-cell LPDs primary to the spleen, commonly presenting with splenomegaly and co-involvement of peripheral blood and bone marrow, but with little to no involvement of lymph nodes. Splenic marginal zone lymphoma is one of the prototypical, best studied, and most frequently encountered CD5-CD10-LPD of the spleen and typically involves white pulp. In contrast, hairy cell leukemia, another well-studied CD5-CD10-LPD of the spleen, involves red pulp, as do the two less common entities comprising so-called splenic B-cell lymphoma/leukemia unclassifiable: splenic diffuse red pulp small B-cell lymphoma and hairy cell leukemia variant. Although not always encountered in the spleen, lymphoplasmacytic lymphoma, a B-cell lymphoproliferative disorder consisting of a dual population of both clonal B-cells and plasma cells and the frequent presence of the MYD88 L265P mutation, is another CD5-CD10-LPD that can be seen in the spleen. Distinction of these different entities is possible through careful evaluation of morphologic, immunophenotypic, cytogenetic, and molecular features, as well as peripheral blood and bone marrow specimens. A firm understanding of this group of low-grade B-cell lymphoproliferative disorders is necessary for accurate diagnosis leading to optimal patient management.

**Keywords:** spleen; splenic marginal zone lymphoma; hairy cell leukemia; hairy cell leukemia variant; lymphoplasmacytic lymphoma; splenic diffuse red pulp small B-cell lymphoma





## 1. Introduction

The spleen is a curious and often understudied organ. Many clinical entities primary to the spleen have only been published in a relatively small series of cases. B-cell lymphoproliferative disorders, along with vascular tumors, are some of the most common neoplasms of the spleen and well-characterized neoplasms of the spleen. Lymphomas of the spleen vary from small B-cell lymphomas such as splenic marginal zone lymphoma to large B-cell lymphomas and even T-cell lymphomas, the most common type presenting in the spleen being hepatosplenic T-cell lymphoma. In Falla et al.'s study of 6450 patients with a variety of splenic lymphomas, 48% were splenic marginal zone lymphoma (SMZL), 27% diffuse large B-cell lymphoma (DLBCL), 5% follicular lymphoma (FL), 4% mantle cell lymphoma (MCL), and 4% were T-cell lymphoma [1].

Low-grade B-cell lymphoproliferative disorders (LPDs) are by far the most frequent type of LPDs of the spleen. These lymphomas can be primary to the spleen, and they can secondarily involve the spleen after being detected in another anatomic location. Primary splenic lymphomas have involvement limited to the spleen or can reside in hilar lymph nodes next to the spleen, but usually do not involve other lymph nodes. Primary splenic lymphomas are usually non-Hodgkin B-cell lymphomas that most commonly present

as splenomegaly. These primary spleen lymphomas often infiltrate the bone marrow as well as show a leukemic component in the blood. Secondary lymphomas of the spleen include DLBCLs as well as other small B-cell lymphomas such as chronic lymphocytic leukemia/small lymphocytic lymphoma (CLL/SLL), FL, MCL, and lymphoplasmacytic lymphoma (LPL), although MCL, FL, and LPL may instead present as primary to the spleen in rare cases. In fact, a variety of low-grade B-cell LPDs can involve the spleen, and accurate classification of these lymphomas can be difficult [2].

Whereas splenectomy used to be the only source of splenic tissue for diagnosis, biopsy of the spleen is more commonly being performed, which poses the additional challenge of diagnosing these lymphomas with a limited amount of tissue. Reactive hyperplasia of the spleen can also present with splenomegaly and differentiating between reactive hyperplasia and lymphoma can be difficult in some cases, especially on core biopsies with scant material. A precise diagnosis of splenic lymphoma may be necessary for treatment and may require assessment of the histologic features along with phenotyping by flow cytometry and/or immunohistochemistry; in some cases, it may also require molecular testing using fluorescent in situ hybridization, cytogenetic testing, polymerase chain reaction (PCR) for gene rearrangement studies, and rarely, next-generation sequencing studies.

This paper reviews the CD5-negative, CD10-negative low-grade B-cell lymphoproliferative disorders (CD5-CD10-LPD) of the spleen. These LPDs comprise a fascinating group of indolent, neoplastic, mature B-cell proliferations that are essential to accurately identify but can be difficult to diagnose. They comprise the majority of B-cell LPDs primary to the spleen, and most commonly present with splenomegaly. They can also extend outside of the spleen, and the peripheral blood and bone marrow of these patients are frequently co-involved by the LPD. In fact, features specific to the peripheral blood such as hairy cell projections and bipolar cytoplasmic blebs are essential for diagnosing the disease. There are also features within the bone marrow, such as the fried egg appearance of hairy cell leukemia (HCL), that are helpful in identifying the lymphoma; therefore, blood and bone marrow specimens are also necessary for diagnosing these diseases. There is often little to no involvement of lymph nodes, with the exception of hilar lymph nodes, which are considered to be an extension of the spleen. SMZL is one of the prototypic, best studied, and most frequently encountered CD5-CD10-LPD of the spleen that preferentially involve white pulp. In contrast, HCL involves red pulp, as do the two less common entities comprising so-called splenic B-cell lymphoma/leukemia unclassifiable: splenic diffuse red pulp small B-cell lymphoma (SDRPL) and hairy cell leukemia variant (HCL-v). Although not always encountered in the spleen, LPL, a B-cell LPD consisting of a dual population of both clonal B-cells and plasma cells and the frequent presence of the *MYD88* L265P mutation, is another CD5-CD10-LPD that can be seen in the spleen. Distinction of these different entities is possible through careful evaluation of morphologic, immunophenotypic, and molecular features as well as peripheral blood and bone marrow specimens. A firm understanding of this group of low-grade B-cell LPDs is necessary for accurate diagnosis leading to optimal patient management.

## 2. Splenic Marginal Zone Lymphoma

Splenic marginal zone lymphoma (SMZL) is one of the most common low-grade LPDs to involve the spleen and is often the first lymphoma considered among the CD5-CD10-LPDs. SMZL is a mature B-cell neoplasm composed of small lymphocytes that arise from the spleen and are thought to be specifically derived from the marginal zone. Although it is the most frequent primary splenic lymphoma, SMZL comprises less than 2% of all lymphomas throughout the body. Liu et al., in a study of the incidence of SMZL in the United States from the Surveillance, Epidemiology, and End Results (SEER) program, determined that only 0.6% of registry cases from 2001 to 2008 were SMZL [3].

The World Health Organization (WHO) separates marginal zone lymphoma into SMZL, nodal marginal zone lymphoma (NMZL), and extra-nodal marginal zone lymphoma (ENMZL). NMZL and ENMZL are more common than SMZL. Generally involving adult

men and women in equal numbers, SMZL is usually diagnosed in patients over 50 years of age and is more common in white patients than in patients of other ethnicities. Affected patients typically present with splenomegaly and/or splenic hilar lymphadenopathy, but other extra-nodal lymphadenopathy is only rarely present. Splenomegaly is reported in 75% of patients and B symptoms are only rarely present. Most patients are asymptomatic, although some can present with abdominal pain/discomfort due to splenomegaly. Serum monoclonal paraproteins can be detected in a subset of patients with SMZL. A variety of autoimmune diseases are present in almost 20% of patients with SMZL, including autoimmune hemolytic anemia and Still disease. There is also an association with hepatitis C virus in approximately 20% of the patients in Southern Europe, particularly with Italian patients [4].

### 2.1. Spleen and Lymph Node Involvement

By gross examination and by imaging studies of the spleen, SMZL has a characteristic growth pattern, presenting with numerous miliary white nodules throughout the spleen with a diffuse spread. Table 1 shows the gross manifestations of different spleen neoplasms, splitting them into those presenting with diffuse miliary white nodules, large mass-like nodules, or a homogenous beefy red cut surface. SMZL is the only CD5-CD10-LPD that presents with diffuse miliary white nodules. SLL/CLL, FL, and MCL can present similarly, but they are usually CD5+ or CD10+ and are usually secondary lymphomas to the spleen. HCL and HCL-v do not show diffuse miliary nodules and instead present with a diffuse beefy red enlarged spleen without miliary nodules [5].

**Table 1.** Gross manifestations of splenic neoplasms.

|  | **Low-Grade CD5(−)CD10(−) B-Cell LPDs** | **Other Neoplasms** |
|---|---|---|
| Diffuse miliary small white nodules | Splenic marginal zone lymphoma | Follicular lymphoma<br>Mantle cell lymphoma<br>CLL/SLL<br>Classic Hodgkin lymphoma, rarely |
| Large mass-like nodules, solitary or multiple |  | Diffuse large B-cell lymphoma, follicular lymphoma, usually higher grade, classic Hodgkin lymphoma; Vascular neoplasms including littoral cell angioma, hemangioma, and splenic hamartoma |
| Homogeneous, beefy red cut surface | Hairy cell leukemia, hairy cell leukemia variant, splenic diffuse red pulp small B-cell lymphoma | Hepatosplenic T-cell lymphoma, T-cell prolymphocytic leukemia |

Abbreviations: CLL/SLL = chronic lymphocytic leukemia/small lymphocytic lymphoma; low grade CD5(−)CD10(−) B-cell LPDs = low grade CD5 negative, CD10 negative LPDs.

Thought to be derived from the splenic marginal zone, SMZL can sometimes but not always show distinct marginal zone differentiation. The neoplastic B-cells primarily involve the white pulp with only secondary extension into the red pulp. This is an important point because SMZL is the only one of the CD5-CD10-LPDs focused on the white pulp, since HCL, HCL-v, LPL, and splenic diffuse red pulp small B-cell lymphoma are all diseases centered on red pulp.

In SMZL, the white pulp is typically expanded into scattered large nodules or coalescing aggregates [6]. The white pulp often shows a biphasic appearance with more pale lymphocytes at the outer periphery of the white pulp and darker B-cells at the inner white pulp (Figure 1). As described in the latest iteration of the WHO classification, the small B-cells surround and replace the splenic white pulp germinal centers, efface the follicle mantle, and merge with a peripheral marginal zone of large cells including scattered transformed cells [7]. The nodular infiltrate consists of reactive secondary follicles that are

either encircled, infiltrated, or entirely replaced by the malignant marginal zone B-cells. Some cases of SMZL do not show the typical biphasic pattern, but instead, demonstrate a monomorphic pattern [8]. In these monomorphic cases, the neoplastic marginal zone B-cells have fully replaced the residual germinal centers, so they have a more homogeneous appearance. This monomorphic/monophasic appearance is more commonly seen in advanced disease.

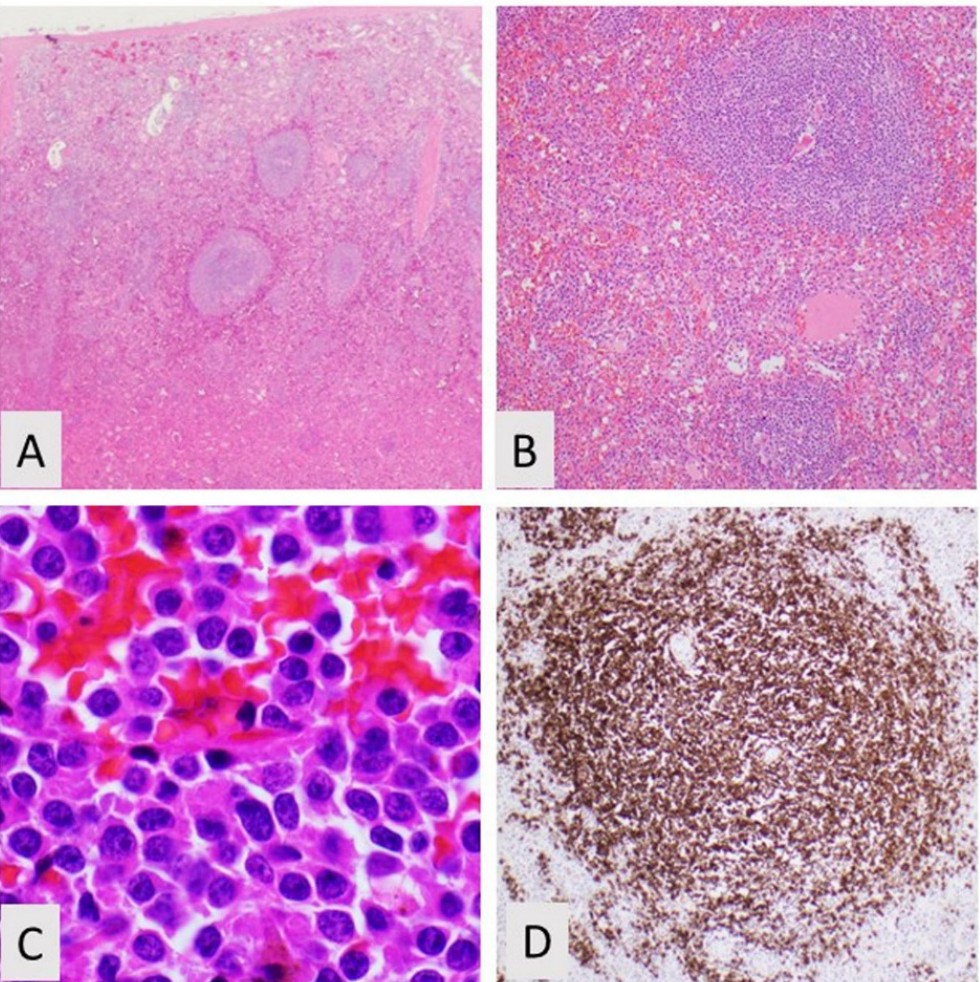

**Figure 1.** Splenic marginal zone lymphoma. (**A**) Low power of spleen with prominent follicles, H&E, 20×; (**B**) Atypical follicle comprised of small lymphocytes, H&E, 100×; (**C**) High power of plasmacytic lymphocytes, H&E, 400×; (**D**) Immunohistochemistry showing CD20 positivity. CD5 and CD10 are negative (not shown).

The cellular morphology of SMZL comprises germinal center cells within the inner darker zone that are mostly small in size with round nuclei, coarse chromatin and scant cytoplasm, and neoplastic marginal zone B-cells in the outer zone that have minimally irregularly shaped nuclei, more vesicular chromatin, and increased pale cytoplasm. Plasmacytoid differentiation with cells having eccentric nuclei, perinuclear hofs, and ample pink cytoplasm has been reported to occur approximately 40% of the time [9]. There are also, typically, scattered larger transformed centroblasts and/or immunoblasts, more commonly in the outer zone. There are cases of SMZL with increased numbers of large, atypical cells in the outer zone, and these cases may present with more aggressive clinical disease [10]. In early manifestations of SMZL, the outer pale cells may be inconspicuous and the inner dark germinal centers are intact, and it can be difficult to distinguish from reactive marginal zone hyperplasia. Mitotic figures are usually rare or inconspicuous.

Sheet-like large aggregates of neoplastic B-cells should not be present, and their presence would be diagnostic for more aggressive lymphoma, with the most common type being diffuse large B-cell lymphoma. Cases of SMZL transforming into diffuse large B-cell lymphoma have been identified, and this transformation connotes a more aggressive disease [11].

Although the white pulp is the primary focus of SMZL, the red pulp is secondarily involved in most cases by neoplastic B-cells infiltrating the splenic cords and sinuses. In early disease, only small clusters of B-cells may be present in the red pulp. In advanced disease, the red pulp can be difficult to identify because it is replaced by sheets of neoplastic B-cells [12]. Small aggregates of epithelioid histiocytes can also be seen in the red pulp.

The hilar lymph nodes near the spleen are usually also involved by SMZL. These hilar lymph nodes may only be partially replaced by SMZL and still have dilated sinuses [13]. Hilar lymph nodes may show marginal zone differentiation with plasmacytoid or monocytoid change and colonization of reactive germinal centers. The hilar lymph nodes are thought to be an extension of the spleen, and hilar lymphoma involvement is not considered to increase the stage of the disease.

### 2.2. Blood and Bone Marrow Morphology

Previously, SMZL was called splenic lymphoma with circulating villous lymphocytes (SLCVL). Although SLCVL is not a currently used term, it is a descriptive term that brings to attention the abnormal lymphocytes that are frequently seen in the peripheral blood in SMZL. In 1987, Melo et al. first described patients with splenomegaly who also presented with abnormal circulating lymphocytes and an absolute lymphocytosis [14]. These lymphocytes with characteristically thin, short, and unevenly distributed villi were subsequently proven to be monoclonal. Blood and bone marrow are thought to be involved in more than 80% of SMZL patients, and an absolute lymphocytosis has been identified in approximately 60% of patients. Leukopenia can also be present in some cases, and anemia and thrombocytopenia have been noted in a subset of cases. Anemia and thrombocytopenia are thought to be a result of hypersplenism and, less commonly, from the replacement of the bone marrow [15,16]. A subset of these cases with blood involvement present as monoclonal B-cell lymphocytosis without splenomegaly. Only a small fraction of patients with monoclonal B-cell lymphocytosis develop SMZL [17]. In summary, the neoplastic B-cells in the peripheral blood, so-called villous lymphocytes, have unipolar short cytoplasmic projections and are considered a part of the SMZL disease.

The bone marrow can also be involved in SMZL. In the bone marrow, neoplastic involvement with SMZL presents with increased numbers of B-cells. These B-cells more characteristically show an intrasinusoidal or perivascular growth pattern as identified in approximately 40% of cases [18]. Less frequently, the neoplastic B-cells can be present in nodular, paratrabecular, diffuse, or mixed patterns. Histologically, the B-cells in the bone marrow look identical to those in the spleen.

### 2.3. Phenotyping by Immunohistochemistry and Flow Cytometry

Immunophenotyping by flow cytometry and/or by immunohistochemistry is essential to separate SMZL from the other CD5-CD10-LPDs of the spleen. The SMZL B-cells may have a somewhat nonspecific phenotype, but this phenotype can still be used to distinguish it from other LPDs such as HCL, which expresses more specific markers. Using flow cytometry, the neoplastic cells of SMZL are routinely positive for all pan-B-cell markers, such as PAX5 and CD20, and show immunoglobulin light chain restriction by kappa or lambda. The heavy chains are usually IgM positive as well as IgD positive. This differentiates SMZL from NMZL and ENMZL, which usually do not express IgD. Other markers immunoreactive for SMZL include FMC7, CD27, and CD200, which may be dimly positive [19]. CD10 is negative and CD5 is usually negative, although it can be positive in approximately 20% of cases. In this minority of CD5+ cases, other entities such as MCL and CLL/SLL would also need to be excluded. CD5+ SMZL has a similar prognosis to the

more common CD5− SMZL, although studies have shown that CD5+ SMZL presents with a higher blood lymphocytosis and a larger B-cell bone marrow infiltrate [20]. The lack of CD23 and LEF1 help exclude CLL/SLL, and the absence of cyclin D1 and SOX11 help to rule out MCL.

Some HCL markers can be expressed in marginal zone lymphoma. CD11c is positive in approximately 50% of SMZL cases. CD25 is also reactive in approximately 30% of cases, and CD103 can rarely be positive. These markers are often only partially or weakly positive, and in a given SMZL case usually only one of the markers is expressed. This distinguishes SMZL from HCL, which often strongly and diffusely expresses all three markers (CD11c, CD25, and CD103).

By immunohistochemistry, other B-cell markers, such as CD79a and OCT2, are also expressed. CD23 can be rarely expressed, but cyclin D1 and SOX11 should be negative. Other HCL markers, including annexinA1, CD123, and BRAF, should be negative. CD10 and BCL6 are negative in the neoplastic B-cells but will be positive in any residual reactive germinal center. These germinal centers will have underlying follicular dendritic cell meshworks that are highlighted with immunohistochemistry for CD21, CD23, and CD35. Moreover, up to half of the cases will express CD21 and CD35 on the B-cells. The proliferative marker ki-67 is said to often show a targetoid pattern in SMZL where there is increased staining in the central reactive germinal centers as well as increased proliferation in bands in the outer marginal zones. Table 2 lists different immunohistochemical markers in SMZL as well as the different LPDs in its differential diagnosis to provide more details about these stains.

**Table 2.** Phenotyping splenic marginal zone lymphoma and its differential diagnosis.

| Antibody | Immunoreactivity | Staining Pattern | Description | Also Detected in |
|---|---|---|---|---|
| Annexin A1 | No | Cell membrane | HCL, specific marker | HCL |
| BRAF | No | Nuclear | None | HCL |
| Cyclin D1 | No | Nuclear | None | MCL, PCM |
| BCL2 | Yes | Cell membrane and cytoplasm | (+) in neoplastic B- cells, (−) in residual germinal centers | Most other small B-cell lymphomas |
| BCL6 | No | Nuclear | (−) in neoplastic cells, (+) in residual germinal centers, may be (+) in transformed SMZL | BL, DLBCL, FL |
| CD5 | Yes | Cell membrane | (+) in ~20% of cases, often dim reactivity | CLL/SLL, MCL |
| CD10 | No | Cell membrane | (+) only in residual germinal centers | BL, DLBCL, FL |
| CD11c | Yes | Cell membrane | Flow marker, (+) in 50% of cases | HCL |
| CD20 | Yes | Cell membrane | Pan-B-cell marker, neoplastic B-cells (+) | Most other B-cell lymphomas |
| CD21 | Yes | Dendritic pattern | (+) in follicular dendritic cell meshworks | FL |

**Table 2.** *Cont.*

| Antibody | Immunoreactivity | Staining Pattern | Description | Also Detected in |
|---|---|---|---|---|
| CD23 | No | Dendritic pattern | (+) in follicular dendritic cell meshworks | CLL/SLL, FL |
| CD25 | Yes | Cell membrane | Flow and immunohistochemistry marker, (+) in subset of cases | HCL |
| CD103 | Yes | Cell membrane | Flow and immunohistochemistry marker, (+) in small % of cases | HCL |
| DBA44 | Yes | Cell membrane | (+) in subset of cases | HCL |
| IgD | Yes | Cell membrane | (+) in subset of cases | Reactive mantle zones |
| IgM | Yes | Cell membrane | (+), most cases | |
| Ki-67 | Yes | Nuclear | Positive staining in both the germinal center and marginal zone which has been described as the so-called targetoid pattern. | Differing proliferation index in each lymphoma |

Abbreviations: BL = Burkitt lymphoma, CLL/SLL = chronic lymphocytic leukemia/small lymphocytic lymphoma, DLBCL = diffuse large B-cell lymphoma, FL = follicular lymphoma, HCL = hairy cell leukemia, MCL = mantle cell lymphoma, PCM = plasma cell myeloma.

### 2.4. Molecular Testing of SMZL

Molecular testing can be helpful in establishing an SMZL diagnosis and in excluding some of the other entities in the differential diagnosis. As SMZL is a B-cell lymphoma, PCR is useful to show that there is clonal rearrangement of the immunoglobulin heavy and light chain genes. These genes should not typically be rearranged in reactive marginal zone hyperplasia, which is often in the differential diagnosis, especially in early cases of SMZL. Approximately 50% of SMZL cases show somatic hypermutation of immunoglobulin variable region genes, although this has not been proven to correlate with prognosis. SMZL has a bias towards using *IGVH1-2\*04*, and *IGVH1-2\*04* and is thought to be associated with 7q deletions and 14q alterations. The predilection of SMZL for *IGVH1-2\*04* helps separate it from the other CD5-CD10-LPDs, including HCL, HCL-v, and splenic diffuse red pulp small B-cell lymphoma, which do not preferentially express *IGVH1-2\*04*.

SMZL does not show the same translocations detected in other small B-cell lymphomas. In particular, the translocations detected in extranodal EMZL of mucosal-associated lymphoid tissue (MALT), including t(11;18)(q21;q21) involving *BIRC3/MALT1*, t(1;14)(p22;q21), and t(3;14)(p14;q32) are not found in SMZL. Furthermore, *BRAF* mutations, which are often considered diagnostic in HCL, are not identified in SMZL. *MYD88* mutations, the hallmark of LPL, are only detected in approximately 10% of SMZL. *MYD88* gene mutations are used in conjunction with morphology and phenotype to distinguish SMZL from LPL.

Although SMZL does not typically demonstrate the recurrent translocations seen in other small B-cell lymphomas, other molecular abnormalities can be detected, especially by cytogenetic karyotyping. Cytogenetic abnormalities by classic karyotyping are detected in approximately 75% of SMZL cases, and complex karyotypes can be identified in almost half of the cases [21]. Chromosome 7 abnormalities are most frequent, followed by changes in chromosomes 1, 3, 6, 8, 12, and 14. The most frequent structural change is the deletion of 7q, which is detected in approximately 40% of SMZL cases. Gains in chromosome 3/3q is the second most common, in approximately 25% of cases. There are other recurrent abnormalities, and these include gains in 8q, 9q, 12q, and 18q along with losses of 6q, 8p, 14q, and 17p, which are seen in up to 20% of cases. Del 7q is the sole cytogenetic abnormality in up to 25% of cases, indicating that it is an essential and early abnormality in SMZL. The deletion of 7q has different breakpoints, with 7q22 being the most common. Del 7q, although appearing to be important for SMZL, can also be seen in some of the other small B-cell lymphomas, and has also been detected in fewer than 5% of MALT

lymphomas [22]. The *CDK6* gene at 7q21 has been seen in chromosome translocations with the *IGH* gene, which yields the t(2;7)(p12;q21).

SMZL does not have a recurrent diagnostic and specific gene alteration; however, some genes have been seen to show alterations. Identifying these molecular subgroups of SMZL may help for disease classification, its treatment, and understanding its pathogenesis [23]. *NOTCH2* and *KLF2* are two of the most frequently altered genes. *NOTCH2* is a gene required for marginal zone development, and *NOTCH2* gain-of-function somatic mutations have been identified in approximately 25% of patients [24].

The *NOTCH2* mutations are not found in other small B-cell lymphomas or HCL, which would be in the differential diagnosis of SMZL [25]. Rare cases of diffuse large B-cell lymphoma (8% of cases) and ENMZL have shown *NOTCH2* mutations [26]. *NOTCH2* mutations can be used to differentiate SMZL from other small B-cell lymphomas in difficult cases or where tissue is limited. *KLF2,* or the Kruppel-like factor 2 zinc finger gene, has been detected in approximately 30% of SMZL cases and is thought to stop the suppression of the NF-kB pathway [27]. *KLF2* mutations can be detected in a low percentage of other small B-cell lymphomas and are not specific for SMZL. Other genes that appear to be recurrently altered in SMZL include *TP53*, *TNFAIP3*, *KMT2D*, *MYD88*, *TRAF3*, and *SPEN* [28]. By gene expression profiling, B-cell receptor signaling pathways and activation of *AKT1* have been implicated in SMZL. Its miRNA profile shows overexpression of microRNAs including miR-155, miR-21, and others.

*2.5. Treatment and Prognosis*

SMZL is often an indolent disease with a good prognosis and slow clinical course. The overall median survival is at least 10 years, and asymptomatic patients may be followed without receiving chemotherapy treatment. Splenectomy or rituximab is often utilized, especially if the patients have cytopenias. Splenectomy is the oldest treatment and was traditionally used first to alleviate any splenomegaly symptoms and cytopenias. Splenectomy is still sometimes performed for diagnosis and not for treatment, but it does not have to be an initial therapy for SMZL. SMZL is a systemic disease involving the spleen, bone marrow, and peripheral blood, and splenectomy will not help mitigate any disease in the blood or the bone marrow. Instead, diagnosis can be performed by splenic biopsy along with phenotyping by flow cytometry and immunohistochemistry. Then, splenectomy can be reserved as palliative therapy for patients who do not respond to other immunotherapy such as rituximab [29,30].

SMZL is incredibly responsive to rituximab. The overall response rate is greater than 90%, and the 5-year progression-free survival is approximately 70% [31]. SMZL patients generally do not have a good response to chemotherapy used for other low-grade B-cell lymphomas. SMZL patients that also have HCV infection may be treated with interferon-γ or ribavirin. Low dose radiation is usually only a second-line therapy for those with splenomegaly who failed other treatments [32]. A small portion (approximately 15%) of SMZL patients may have more aggressive disease. It is usually only when SMZL patients develop transformation to a higher-grade lymphoma that they experience an aggressive clinical course [33]. One in ten cases is thought to transform into a large B-cell lymphoma. Prolymphocytic leukemia transformation is rare but generally connotes a poorer prognosis. Unfavorable prognostic factors include large tumor size, *NOTCH2* mutations, *TP53* mutations, 7q deletion, and unmutated *IGH* variable chain region genes.

Newer therapeutic agents have been extensively studied in either nodal marginal zone or extranodal marginal zone lymphoma, but much less so in splenic marginal zone lymphoma. The combination of bendamustine and rituximab has never been tested in a dedicated clinical trial for the splenic marginal zone. However, this combination has been used in non-Hodgkin lymphoma and is very effective in other types of marginal zone lymphoma. B cell receptor inhibitors such as ibrutinib and immunomodulatory agents such as lenalidomide have been FDA approved in marginal zone lymphoma, but not specifically

in splenic marginal zone lymphoma. PI3K inhibitors are similarly effective in non-Hodgkin lymphoma including marginal zone lymphoma.

## 3. Hairy Cell Leukemia

### 3.1. Introduction

Hairy cell leukemia (HCL) is a well-characterized but uncommon indolent neoplasm of mature B-cells with distinct clinical, morphologic, immunophenotypic, and molecular features. It typically presents as a leukemia involving the spleen, liver, bone marrow, and peripheral blood without lymph node involvement. Correct diagnosis of HCL is important, as treatment is well established, usually with good effect.

### 3.2. Biological Features/Pathogenesis

HCL was first reported in the literature in 1958 in a manuscript by Bouroncle and colleagues who described a neoplasm of malignant "hairy cells" with a distinct propensity for infiltrating lymphoreticular tissue, in particular spleen, liver, and bone marrow [34]. While the condition was originally termed "leukemic reticuloendothelioisis," subsequent studies concentrating on the light microscopic and ultrastructural features of the malignant cells emphasized their striking and distinctive hairlike cytoplasmic projections, which resulted in the gradual replacement of the now obsolete term "leukemic reticuloendothelioisis" with the current and more memorable name "hairy cell leukemia". Advances in immunological methods in the 1970s and 1980s resulted in the discovery that "hairy cells" are B-cells demonstrating surface expression of pan-B-cell markers such as CD19, CD20, and CD22 as well as light chain-restricted surface immunoglobulin receptors indicative of monoclonality. Further investigation into the nature of "hairy cells" revealed up-regulation of antigens associated with B-cell activation including FMC7, CD25, CD72, CD40L, CD11c, CD103, BCL, and TRAP; down-regulation of antigens often reduced during activation including CD21 and CD24; absence of germinal center antigens such as CD10 and BCL6; and absence of plasma cell antigens including CD138 and MUM1. Collectively, these immunophenotypic studies suggested that HCL is derived from an activated mature B-cell; however, it would not be until later that more sophisticated molecular techniques confirmed and expanded this notion (see below) [35].

Investigations into adhesion molecule expression by HCL B-cells helped elucidate some of the reasons for the unique lymphoreticular distribution of HCL as well other features of the disease such as bone marrow fibrosis. HCL B-cells were found to constitutively express various adhesion molecules, including a number of different integrins as well as CD44. Integrin expression results in HCL B-cells binding to several extracellular matrix elements, including fibronectin and vitronectin, whereas CD44 expression allows for the adherence to hyaluronan. Bone marrow is rich in hyaluronan, which helps account for the bone marrow tropism seen in HCL. In bone marrow, interaction with hyaluronan stimulates autocrine production of fibroblast growth factor (FGF) by HCL B-cells, which in turn stimulates fibronectin production by the same cells. Fibronectin is an important constituent in the bone marrow fibrosis seen in HCL, and its interstitial deposition in HCL coupled with HCL B-cells' strong adherence to fibronectin likely help explain the unique interstitial growth pattern of HCL in bone marrow (see Clinicopathologic and Diagnostic Features, below). In the spleens of patients with HCL, fibronectin accumulation (and consequently fibrosis) is not seen because hyaluronan is absent. Vitronectin, in contrast, is abundant in the red pulp of the spleen, and it is likely the interaction of HCL B-cells and vitronectin through specific integrin proteins that results in the unique red pulp distribution of HCL in the spleen (see Clinicopathologic and Diagnostic Features, below). Another unique splenic feature of HCL likely attributable to integrin expression is the pseudosinus ("blood lake") formation seen in the red pulp (see Clinicopathologic and Diagnostic Features, below). Such pseudosinus formation requires vascular remodeling, which in HCL likely stems from integrin-mediated interaction of HCL B-cells with vascular endothelial cells,

resulting in replacement of endothelial cells by HCL B-cells and subsequent "blood lake" formation [35,36].

In addition to adhesion molecule expression, studies on cytokine expression by HCL B-cells have provided further insight into the pathogenesis of HCL. In addition to the aforementioned FGF, a number of other cytokines have been shown to be constitutively produced and released by HCL B-cells. These include TNF-α, IL-6, and TGFβ, all of which contribute to the disease phenotype of HCL. In the bone marrow compartment, TGFβ has two main effects. The first is inhibition of normal hematopoiesis, which helps explain the pancytopenia that typically accompanies HCL (see Clinicopathologic and Diagnostic Features, below). The second is stimulation of bone marrow fibroblasts to produce the reticulin fibers involved in the bone marrow fibrosis of HCL. While TGFβ's effects in HCL stem from paracrine interactions with nearby marrow constituents, the effects of TNF-α and IL-6 are due to autocrine stimulation of cognate receptors expressed on the surface of the HCL B-cells (similar to FGF). Autocrine stimulation of HCL B-cells by TNF-α and IL-6 results in inhibition of apoptosis, possibly through overexpression of BCL2, resulting in prolonged cell survival and maintenance of the neoplastic cell population [35,36].

Investigations into the molecular genetics of HCL in recent years have added greatly to our understanding of the biology of this disease, with significant implications for both diagnosis and treatment. Molecular studies of antigen receptor genes in HCL show monoclonal *IGH* gene rearrangements in almost all cases, consistent with a neoplastic B-cell process. In over 85% of cases of HCL, the IGHV genes demonstrate somatic hypermutation, which is consistent with a post-germinal center stage of maturation [35,37]. DNA microarray analysis showed that HCL exhibits a unique gene expression profile not seen in other B-cell neoplasms. This analysis found that HCL B-cells have a gene expression pattern most similar to memory B-cells, which correlates with earlier immunophenotypic studies suggesting HCL is derived from activated mature B-cells (discussed above), as well as the IGHV somatic hypermutation data indicating a post-germinal center stage of maturation for HCL. The gene expression profiling data also showed that HCL B-cells downregulate genes for certain chemokine receptors important in homing to lymph nodes, including CCR7 and CXCR5, and overexpress certain genes involved in inhibiting matrix metalloproteases (MMPs) including TIMP-1 and TIMP-4. The former feature helps explain the lack of lymph node involvement by HCL, whereas the latter feature helps account for the lack of tissue invasion by HCL, which likely contributes to its unique "leukemic" tissue distribution. Other genes found to be overexpressed in HCL by DNA microarray analysis include β-actin and *GAS7*, which likely play a role in maintaining the unique "hairy cell" morphology of HCL B-cells; *bFGF* and *FGFR1*, which play a role in HCL-mediated bone marrow fibrosis (discussed above); and annexin A1, which likely plays a role in phagocytosis by HCL B-cells and also serves as an important diagnostic marker (discussed below) [35,37–39].

A key insight into the molecular pathogenesis of HCL came in 2011 with the publication of a study detailing the results of whole-exome sequencing of 47 HCL samples. In this study, all of the samples (100%) were found to harbor the *BRAF* V600E mutation [40]. Multiple subsequent studies confirmed a high frequency of *BRAF* V600E mutation in cases of HCL, strongly suggesting this mutation is a disease-defining genetic event important in the pathogenesis of the disease [41–43]. Investigations into the role of mutated *BRAF* V600E in HCL found constitutive activation of the RAF-MEK-ERK mitogen-activated protein kinase (MAPK) pathway secondary to unregulated phosphorylation of the BRAF kinase targets MEK and ERK. This activity was found to prolong cell survival of HCL B-cells as well as induce expression of two markers important in the diagnosis of HCL: CD25 and tartrate-resistant acid phosphatase (TRAP). *BRAF* V600E was also found to contribute to the "hairy cell" morphology of HCL by inducing overexpression and constitutive activation of RHO GTPases and upregulating GAS7 [40,43].

### 3.3. Clinicopathologic Features and Diagnosis

Although well-characterized biologically, HCL is a rare disease with the annual incidence in the United States estimated to be 3.2 cases per every 1,000,000 individuals. It accounts for approximately 2% of all lymphoid leukemias and typically occurs in middle-aged to elderly white men (male:female ratio 4:1) with a median age at diagnosis of 58 years. Diagnosis in the young adult years is very uncommon, but not unheard of. Pediatric cases are exceptionally rare [44].

The clinical presentation of HCL stems from the biological features discussed above, with the most common signs and symptoms secondary to pancytopenia and massive splenomegaly. The massive splenomegaly typically causes left upper quadrant pain, while pancytopenia leads to weakness and fatigue due to anemia, bleeding due to thrombocytopenia, and recurrent opportunistic infections and fever due to leukopenia. Hepatomegaly is also commonly seen, whereas vasculitis, bleeding disorders, neurologic disorders, and immune dysfunction are infrequent. A unique and characteristic feature of the leukopenia seen in HCL is severe monocytopenia. While most cases of HCL present with leukopenia, unusual presentations with normal white blood cell counts and even leukocytosis have been reported [45,46].

Diagnosis of HCL is usually consequent to the pancytopenia seen in most cases, which typically leads to pathologic and laboratory investigation of both peripheral blood and bone marrow samples. The classic finding seen on Wright–Giemsa-stained peripheral blood smears in HCL is the presence of circulating "hairy cells". These are small- to medium-sized lymphocytes exhibiting moderate amounts of pale blue cytoplasm, round or indented nuclei, homogenous spongy chromatin, and inconspicuous or absent nucleoli. The cytoplasm is most notable for thin hairlike projections distributed circumferentially around the cell, which imparts the "hairy cell" morphology of HCL. Occasional small cytoplasmic vacuoles or rod-shaped inclusions can also be seen. The number of circulating "hairy cells" in the peripheral blood can be variable but is usually low [47].

The bone marrow biopsy findings in HCL are characteristic (Figure 2). The most common growth pattern is interstitial or patchy with variable preservation of normal marrow elements depending on the extent of involvement. On hematoxylin and eosin stain, the infiltrate consists of widely spaced lymphocytes demonstrating abundant clear cytoplasm and well-defined cell borders resulting in a so-called "fried-egg" appearance. The cells also exhibit round or indented nuclei, condensed chromatin, and inconspicuous or absent nucleoli. Discrete lymphoid aggregates, often seen in other indolent B-cell LPDs involving bone marrow, are not a feature. The cellularity of the marrow in HCL can vary from hypercellular to hypocellular. An increase in reticulin fibers resulting in bone marrow fibrosis is consistently seen in HCL and can be highlighted by a reticulin stain. Bone marrow fibrosis typically seen in HCL usually results in an inaspirable "dry tap" on attempted bone marrow aspiration. As a consequence, the findings seen on Wright–Giemsa-stained aspirate smears are usually similar to those seen on peripheral blood smears [47].

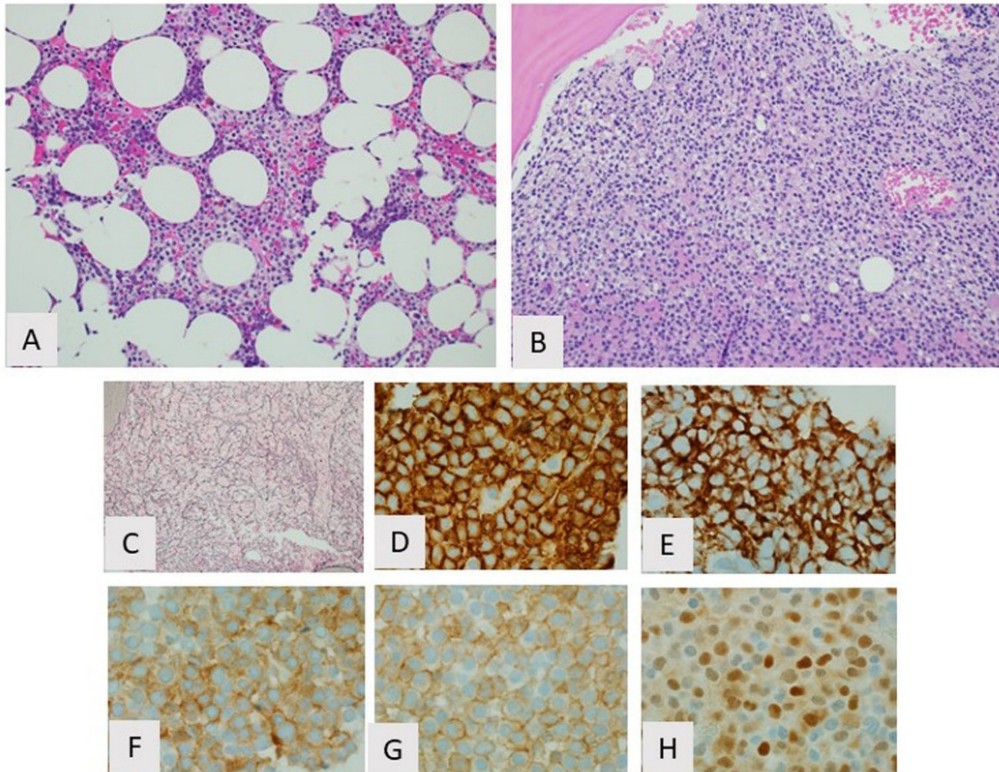

**Figure 2.** Hairy cell leukemia, bone marrow. (**A**) Partial involvement showing typical interstitial growth pattern with "fried egg" appearance, H&E, 20×; (**B**) Diffuse involvement with sheet-like growth pattern H&E, 20×; (**C**) Increased reticulin fibrosis, reticulin stain, 20×. By immunohistochemistry the cells are positive for (**D**) CD20, (**E**) CD103 (**F**) CD25, (**G**) annexin A1, and (**H**) cyclin D1.

While peripheral blood and bone marrow are the most common specimens leading to a diagnosis of HCL, other organs, in particular spleens, are involved by HCL and may be the first tissue type sampled resulting in diagnosis. The spleens of patients with HCL are typically massively enlarged secondary to the gradual accumulation of HCL B-cells as the disease progresses. Grossly, the spleens display a beefy red cut surface with diffused expansion of the red pulp and an indiscernible white pulp. Numerous variably sized "blood lakes" can also be seen grossly. This gross appearance correlates with the histologic findings of HCL in the spleen where the neoplastic B-cells, which exhibit the same "fried-egg" appearance seen in bone marrow, fill the red pulp cords with scattered collections of pooled erythrocytes surrounded by elongated "hairy cells," resulting in "blood lakes". The white pulp is consistently uninvolved and usually atrophic (Figure 3). Aside from peripheral blood, bone marrow, and spleen, the liver is another organ that can harbor HCL, where the "hairy cells" typically occupy the hepatic sinusoids. Involvement of other tissue types such as lymph nodes and skin, although rarely reported, is usually not seen in HCL [47].

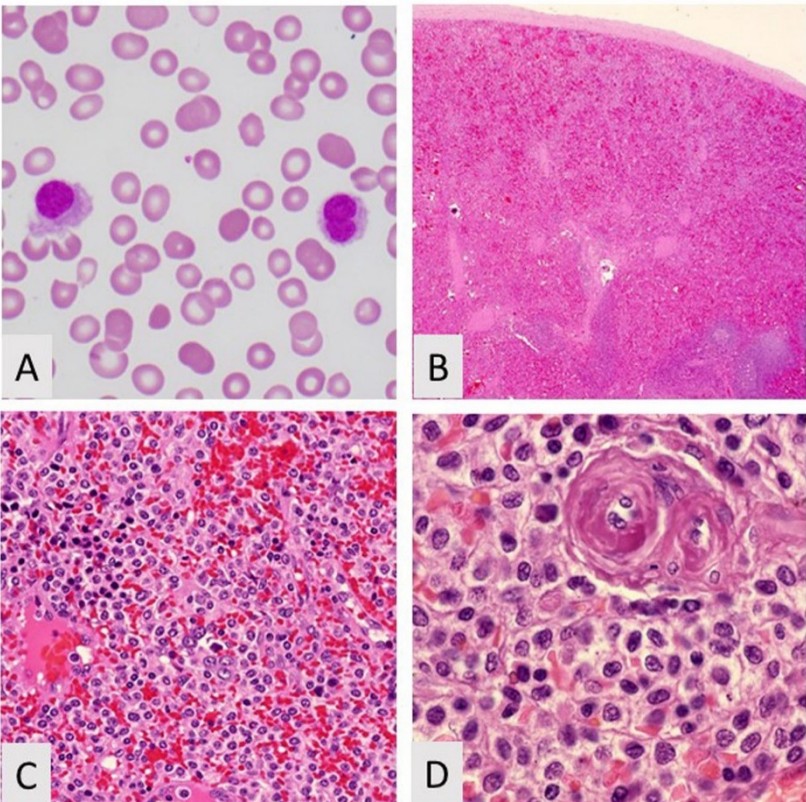

**Figure 3.** Hairy cell leukemia. (**A**) Peripheral blood showing hairy cell, Wright–Giemsa, 400×; (**B**) Low power of spleen showing blood lakes H&R, 20×; (**C**) Intermediate power showing hairy cell leukemia and nucleated red blood cells, H&E, 200×; (**D**) Hairy cell leukemia showing cells with abundant cytoplasm, H&E, 400×.

Although the morphologic features of HCL in peripheral blood, bone marrow, and spleen are characteristic, they can overlap with other entities on the differential diagnosis. For example, the "villous lymphocytes" seen in peripheral blood involvement by SMZL and SDRPL and the "hairy cells" of HCL-v) can look very similar to the "hairy cells" of HCL. Similarly, in bone marrow, the histologic appearance of HCL can sometimes overlap with SMZL, HCL-v, and SDRPL. In the spleen, the red pulp distribution of HCL, while not a feature of the white pulp-based SMZL, is shared by both SDRPL and HCL-v.

Given this potential for morphologic overlap with other disease entities, additional techniques are needed to secure a diagnosis of HCL. Fortunately, the immunophenotype of HCL is unique and is often sufficient to confirm a diagnosis. Whether by flow cytometry or immunohistochemistry, HCL classically shows bright expression of monotypic surface immunoglobulin; bright surface expression of CD19, CD20, CD22, and CD11c; and surface expression of CD25, CD103, CD123, CD200, and FMC7. Additional markers expressed in HCL detected by immunohistochemistry only include annexin A1, TBET, and cyclin D1 [39,46,48–52]. TRAP is another marker expressed in HCL; originally detected cyto-chemically, it is now more easily detected via immunohistochemistry [19]. HCL is usually negative for CD5 and CD10; however, aberrant CD10 expression can be seen in 10% to 20% of cases and aberrant CD5 in 0 to 2% of cases. Other rare immunophenotypic variants have also been reported [46,53–55]. Of the main differential diagnostic entities with HCL, HCL-v has the most immunophenotypic overlap with HCL, being positive for CD11c and CD103. HCL-v, however, is typically negative for CD25, CD123, CD200, annexin A1, TRAP, and cyclin D1. The combination of CD25, CD11c, CD103, and CD123 expression in HCL usually suffices in ruling out SMZL and SDRPL.

When immunophenotypic evaluation does not allow for a definitive diagnosis of HCL, molecular analysis can be helpful. The discovery that the vast majority of HCL cases harbor

the *BRAF* V600E mutation and that this mutation is absent in virtually all other indolent B-cell neoplasms makes detection of this mutation a strong indicator of HCL [40–43]. The *BRAF* V600E mutation can be detected with various sequencing and PCR techniques. Although a strong indicator of HCL, detection of the *BRAF* V600E mutation is not yet a mandatory criterion for the diagnosis of HCL, as rare cases otherwise consistent with HCL that use the IGHV4-34 family of immunoglobulin heavy chain and that have *MAP2K1* mutations lack the *BRAF* V600E mutation [56,57]. No specific cytogenetic abnormalities are associated with HCL [35].

### 3.4. Treatment and Prognosis

HCL is unique among hematolymphoid neoplasms in that it is unusually sensitive to both interferon-α and purine analogs such as pentostatin and cladribine [35,58]. Both classes of drugs appear to have a proapoptotic effect on HCL B-cells with interferon-α's proapoptotic effect appearing to involve autocrine TNF-α overexpression by the neoplastic cells [35,58]. Single-agent purine analog treatment is currently the first-line therapy for HCL, and patients receiving one of these drugs often achieve complete and durable remissions [58,59]. Other indolent B-cell neoplasms in the differential diagnosis of HCL are not as responsive to these drugs, making correct diagnosis important. Although purine analogs are frequently successful in treating HCL, as many as 50% of individuals with HCL will eventually relapse, necessitating the detection of minimal residual disease (MRD) after initial treatment. The best way to monitor MRD in HCL after treatment is by flow cytometry using markers specific to the unique immunophenotype of HCL. Alternative methods for detecting MRD include immunohistochemistry on bone marrow biopsies using antibodies specific for TBET or the V600E mutant BRAF protein [49,60]. Annexin A1, although specific for HCL among indolent B-cell neoplasms, is not a useful MRD marker, as it is also expressed in background myeloid cells [39]. In patients who ultimately develop relapsed refractory disease, treatment strategies include salvage chemotherapy combined with anti-CD20 or anti-CD22 immunotherapy [58,59]. BRAF inhibitors have also shown utility in treating relapsed refractory HCL in patients with a confirmed V600E *BRAF* mutation [43,61].

### 4. Hairy Cell Leukemia Variant

#### 4.1. Introduction

Hairy cell leukemia variant (HCL-v) is a provisional WHO diagnostic entity included in the broader classification of splenic B-cell lymphoma/leukemia, unclassifiable. It refers to cases of B-cell LPDs that resemble classic HCL, but with variant clinical, morphologic, immunophenotypic, and molecular features. The distinction of HCL-v from HCL is important, as the treatment strategies commonly successful in HCL are not as effective in HCL-v [62,63].

#### 4.2. Biological Features/Pathogenesis

HCL-v has only been recognized as a provisional diagnostic entity independent of HCL since 2008, so investigations into the unique biological nature of this neoplasm are relatively limited when compared with HCL. It is known that like HCL, HCL-v is a monoclonal B-cell disorder with a similar "leukemic" lymphoreticular distribution involving peripheral blood, bone marrow, spleen, and liver without significant involvement of lymph nodes or other tissues. While the exact reason for this similarity is not clear, it might stem from similar expression patterns of adhesion molecules, chemokine receptors, and MMP inhibitors. Also similar to HCL, immunophenotypic and molecular features suggest the cell of origin of HCL-v is an activated B-cell at a late stage of maturation, although it is not clear if this is a memory B-cell [62,63].

Although there are similarities between HCL-v and HCL, molecular studies have shown clear differences between these two entities. When compared to HCL, HCL-v shows a higher percentage of cases with no somatic hypermutations in IGHV (33% versus 15%),

with the unmutated HCL-v cases having a high frequency of *TP53* mutations, a feature not recognized in HCL [62,64]. HCL-v also shows more preferential usage of the IGHV4-34 gene family than HCL (40% versus 10%), and gene expression profiling has shown more DNA copy number abnormalities in HCL-v with the most common being gains on chromosome 5 and losses on chromosomes 7q and 17p [64,65]. Perhaps the most significant difference between HCL-v and HCL is the complete absence of the *BRAF* V600E mutation in HCL-v, which is present in the vast majority of HCL cases [40,42,57,66]. Although the *BRAF* V600E mutation is not seen in HCL-v, at least 20% of cases show recurrent mutations in the *MAP2K1* gene, which encodes the MEK1 protein, a downstream phosphorylation target of the BRAF kinase. This finding is also seen in the rare *BRAF* V600E-unmutated cases of HCL expressing IGHV4-34 and suggests that activation of the MAPK pathway may play a role in the pathogenesis of HCL-v [56]. Other biological differences between HCL-v and HCL likely include differential expression of certain cytokines like FGF, as evidenced in the lack of significant bone marrow fibrosis in HCL-v compared to HCL.

*4.3. Clinicopathologic Features and Diagnosis*

HCL-v is a very rare disease, being about 10 times less common than HCL. Like HCL, HCL-v typically occurs in middle-aged to elderly individuals; however, the male predominance seen in HCL is not as prominent in HCL-v, where there is only a slight male predominance. Whereas HCL is most commonly seen in white individuals, HCL-v may be more prevalent in Asian populations [58,62,67].

The clinical presentation of HCL-v is similar to that of HCL with some notable differences. Like HCL, HCL-v typically presents with splenomegaly, which can be massive enough to cause left upper quadrant discomfort. Thrombocytopenia and anemia are also fairly common clinical features in HCL-v, which can predispose to bleeding and weakness/fatigue, respectively. Unlike HCL, HCL-v almost always presents with leukocytosis consisting primarily of circulating "hairy cells" with a normal absolute monocyte count [37,62,63].

Similar to HCL, HCL-v is usually diagnosed following pathologic and laboratory investigation of both peripheral blood and bone marrow samples. The classic finding seen on Wright–Giemsa-stained peripheral blood smears in HCL-v is the presence of many circulating "hairy cells", which, in contrast to those seen in HCL, have a wider variability of both cytoplasmic and nuclear morphologic features and are more numerous (Figure 4). While some "hairy cells" in HCL-v will be indistinguishable from those seen in HCL, many will exhibit variant nuclear features including more condensed chromatin and prominent nucleoli similar to prolymphocytes, or more dispersed chromatin with highly convoluted nuclear contours [62,63]. The presence of "hairy cells" demonstrating prolymphocytic nuclear features has caused some to refer to HCL-v as the "prolymphocytic variant of HCL". The cytoplasmic features of HCL-v are also variable. While hairlike cytoplasmic projections are seen in most of the neoplastic cells, these can be shorter and fewer in number than seen in classic HCL with some cells lacking hairlike projections altogether. In addition, instead of the circumferential distribution of the hairlike projections seen in HCL, some of the "hairy cells" of HCL-v may demonstrate a polarized "villous" distribution of these projections more commonly seen in SMZL and SDRPL [62,63].

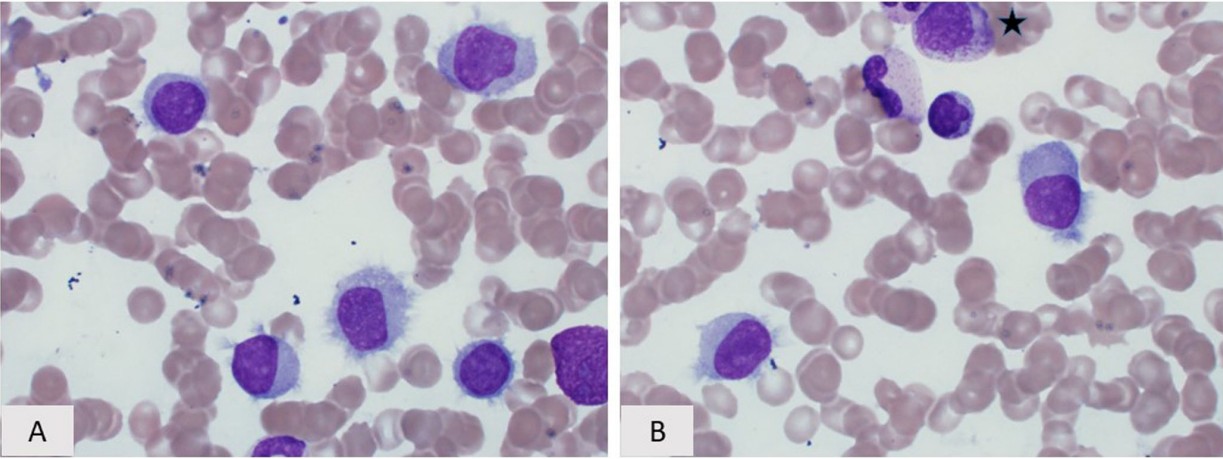

**Figure 4.** Hairy cell leukemia variant. (**A**) Peripheral blood smear and (**B**) Bone marrow aspirate, (a myelocyte is present in the top of the field ★), Wright–Giemsa, 400×. The cytologic features range from typical "hairy cells" showing prominent circumferential cytoplasmic projections, rounded nuclei, spongy chromatin, and inconspicuous/absent nucleoli to cells with less prominent cytoplasmic projections, more finely dispersed chromatin m, and occasional prominent nucleoli.

The pattern of bone marrow involvement by HCL-v is generally very similar to that of HCL; namely, interstitial infiltration with a "fried-egg" appearance. One feature seen in the bone marrows of HCL-v patients, not typically present in HCL, is intrasinusoidal growth, a pattern also associated with SMZL and SDRPL [63]. Unlike HCL, HCL-v does not induce significant reticulin fibrosis so the bone marrow is aspirable [68]. In the spleen, HCL-v diffusely involves and expands red pulp with atrophy of white pulp, similar to both HCL and SDRPL (Figure 5). "Blood lakes" can occasionally be seen, possibly through mechanisms similar to those in HCL. HCL-v also commonly fills and expands red pulp sinusoids, a finding not normally associated with HCL [62,68]. In the liver, HCL-v usually involves both sinusoids and portal tracts.

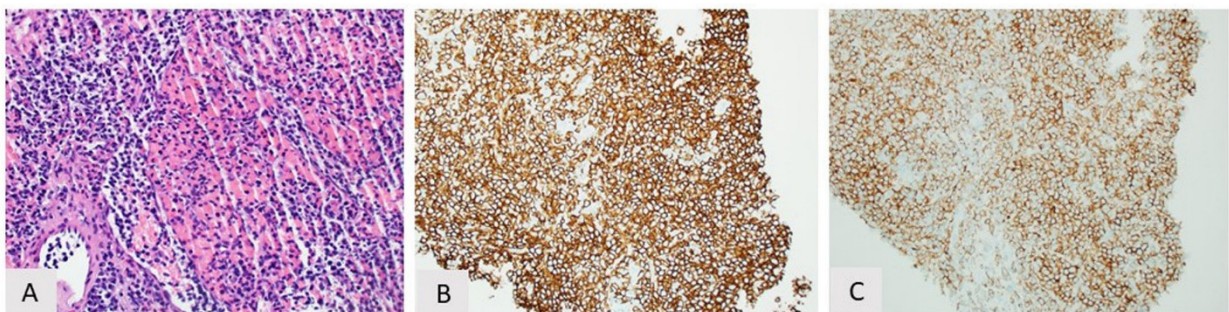

**Figure 5.** Hairy cell leukemia variant, spleen. (**A**) Red pulp involvement with intrasinusoidal growth pattern, H&E, 20×; by immunohistochemistry the cells are positive for (**B**) CD20 and (**C**) CD103, but negative for CD25, annexin A1, CD 123, and BRAF V600E (not shown).

While the variant morphologic features of HCL-v can help distinguish this entity from HCL, immunophenotypic evaluation is necessary for a definitive diagnosis. HCL-v shares several immunophenotypic similarities with HCL; namely, bright expression of monotypic surface immunoglobulin and CD20, as well as co-expression of surface CD11c and CD103 [46,69]. The co-expression of surface CD11c and CD103 helps distinguish HCL-v from SMZL and SDRPL. Importantly, HCL-v is negative for other markers usually expressed in HCL including CD25, CD123, CD200, annexin A1, and TRAP [39,46,54,58,70].

In cases where the immunophenotype does not clearly distinguish HCL-v from HCL, molecular testing for the *BRAF* V600E mutation is helpful, as almost all cases of HCL are positive for this mutation, whereas HCL-v is consistently negative for this change.

Immunohistochemistry targeting the V600E mutant BRAF protein can also be helpful in this regard. No other molecular or cytogenetic features are helpful in the diagnosis of HCL-v.

### 4.4. Treatment and Prognosis

HCL-v tends to have a poorer prognosis than HCL, with an estimated 5-year survival rate of 57%. Recognized adverse prognostic factors include older age, severe anemia, and *TP53* gene mutations [67]. HCL-v, unlike HCL, does not typically respond well to single-agent purine analog treatment, and the optimal therapeutic strategy is not well established [54,58]. Treatment options used in the past include splenectomy and combination chemotherapy with or without anti-CD20 immunotherapy [58]. A recent study documented a good response to combined treatment with the purine analog cladribine and the ant-CD20 agent rituximab [71]. BRAF inhibitors are not useful in HCL-v.

## 5. Lymphoplasmacytic Lymphoma

### 5.1. Introduction

Lymphoplasmacytic lymphoma (LPL) and the corresponding clinical entity Waldenström macroglobulinemia is one of the CD5 negative, CD10 negative small B-cell lymphomas that may involve the spleen. It was first described in the 1940s by Jan Waldenström in a small series of three patients. Since that time, the definition of lymphoplasmacytic lymphoma has greatly been refined. In previous lymphoma classifications, other terms such as well-differentiated lymphocytic plasmacytoid lymphoma, immunocytoma lymphoplasmacytic type, and lymphoplasmacytoid lymphoma have been used to diagnose similar lesions. However, these terms are not exact synonyms for what is now classified as lymphoplasmacytic lymphoma.

In the current WHO classification, lymphoplasmacytic lymphoma is characterized by a proliferation of small mature B lymphocytes, plasmacytoid lymphocytes, and true plasma cells that do not meet criteria for any other small B-cell lymphoma [7]. Lymphoplasmacytic lymphoma and Waldenström macroglobulinemia are sometimes used interchangeably, but they are not the same exact disease. Waldenström macroglobulinemia has a lymphoplasmacytic lymphoma involving the bone marrow or spleen and is associated with an IgM paraprotein. It can involve other tissues including the lymph node, but the lymphoma must involve the marrow. Lymphoplasmacytic lymphoma, on the other hand, is usually associated with a serum monoclonal protein and it is usually IgM, but also rarely be IgG or IgA. Unlike Waldenström macroglobulinemia, lymphoplasmacytic lymphoma does not have to have a monoclonal paraprotein, although the presence of one is helpful for diagnosis. When present, the paraprotein is typically greater than 3 g/dL and can lead to increases in serum viscosity and the clinical manifestations of increased serum viscosity. Notably, patients with LPL/WM characteristically have an IgM monoclonal paraprotein at levels not seen in other small B-cell lymphomas of the spleen.

Clinically, patients are in the seventh decade and often present with symptoms related to hyperviscosity and cytopenias due to bone marrow replacement [72,73].

### 5.2. Morphology and Immunophenotype

Up to one-third of patient present with peripheral blood involvement with circulating lymphocytes, plasmacytoid lymphocytes and even circulating plasma cells. Commonly, rouleaux is also found and is not limited to cases of plasma cell myeloma.

In the bone marrow, LPL/WM shares morphologic features with SMZL making the distinction in this site problematic. LPL may demonstrate diffuse, interstitial, or nodular involvement by a spectrum of lymphocytes, plasmacytoid lymphocytes, and true plasma cells. In the spleen, however, LPL/WM involves the red pulp cords and sinuses, leaving the white pulp atrophic. The infiltrate is again a mixed morphologic infiltrate of mature lymphocytes, plasmacytoid lymphocytes, and plasma cells. SMZL involves the white pulp sparing the red pulp [7]. As a result, on examination of the spleen, the differential diagnosis

for LPL/WM shifts to small B-cell neoplasms involving the red pulp including splenic diffuse red pulp small B-cell lymphoma, HCL, and HCL-v.

### 5.3. Immunophenotype

LPL/WM shares immunophenotypic findings that overlap those of other CD5 negative, CD10 negative small B-cell lymphomas (Figure 6). Characteristically, LPL/WM expresses CD20 and CD79a and is generally negative for CD5 and CD10 [72,73]. CD38 is typically strongly expressed in LPL/WM and the plasma cells in this disorder often co-express CD45 and CD19 [74]. CD138 will stain plasma cells within the neoplasm and light chain restriction is usually demonstrated in plasma cells and often in plasmacytoid lymphocytes as well. LPL/WM also shares immunophenotypic overlap with HCL in that it may express CD25. However, LPL/WM does not express other HCL antigens such as CD11c, CD103, and CD123.

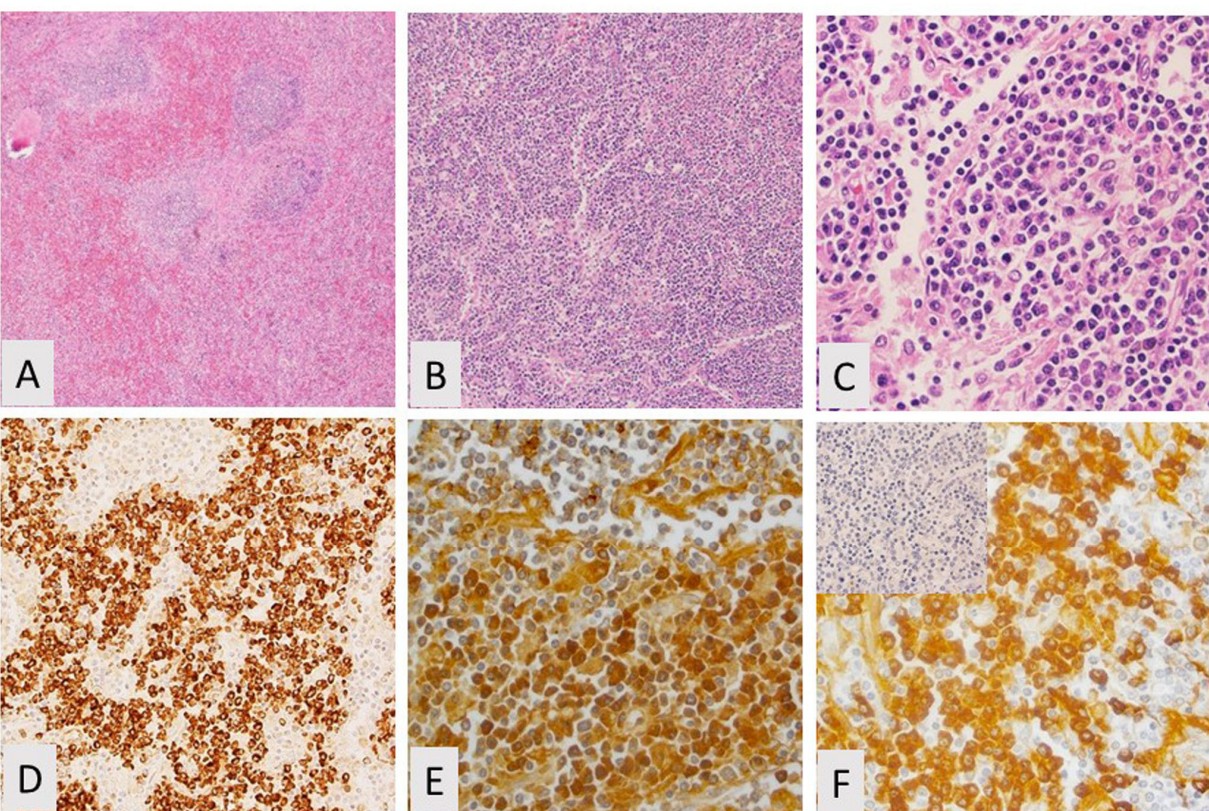

**Figure 6.** Lymphoplasmacytic lymphoma. (**A**) Splenic lymphopasmacytic lymphoma showing disrupted architecture, H&E, 20×; (**B**) Focally showing a more diffuse pattern, H&E, 20×; (**C**) Lymphocytes with a plasmacytic appearance, H&E, 200×; (**D**) Immunohistochemistry (IHC) for CD138 is positive; (**E**) IHC is positive for CD20; (**F**) IHC for kappa is positive, lambda (in set) is negative.

### 5.4. Cytogenetic/Molecular Findings

LPL/WM demonstrates a characteristic, but not specific, *MYD88* L265P mutation in more than 90% of cases. Normally, *MYD88* encodes for a protein that activates the MAPK and NF-KB pathways which subsequently cause proliferation and survival of B-cells. The L265P mutation constitutively activates the NF-KB pathway which leads to tumorogenesis [75,76]. Approximately 30% of LPL/WM have mutated *CXCR4* [77]. CXCR4 is a chemokine receptor which when mutated, can prolong stimulation of CXCL2/SDF1, its ligand, increasing adhesion and migration into the bone marrow [78,79]. Cytogenetic studies show del(6q) and t(9;14)(p13;q32) in up to 50% of cases and trisomy 4 in approximately 20% [76,79–81]. Deletion of 6q may lead to the loss of two tumor suppressor genes, *BLIMP1* (6q21) and *TNFAIP3* (6q23). The t(9;14)(p13;q32) has been correlated with plasmacytoid

differentiation. In this translocation, *PAX5* which encodes for B cell-specific activator protein is translocated to the *IGH* gene at 14q32. This leaves the transcription factor intact and deregulates its expression [79,80]. Trisomy 4 is rarely found in other neoplasms of small B-cells [80–82].

As with other low-grade B-cell lymphomas involving the spleen, careful morphologic assessment, a complete immunophenotypic profile, and cytogenetic/molecular studies are necessary to classify the neoplasm.

## 6. Splenic Diffuse Red Pulp Small B-Cell Lymphoma

Splenic diffuse red pulp small B-cell lymphoma (SDRPL) is currently classified as a provisional entity in the 2016 WHO classification of tumors of the hematopoietic and lymphoid tissues under splenic B-cell lymphoma/leukemia, unclassifiable [7]. SDRPL is an uncommon neoplasm of small, mature appearing, monotonous B-cells that diffusely involve the splenic red pulp. Patients most commonly present with splenomegaly, and typically the spleen is massively enlarged. Cytopenias are also common, but patients with SDRPL do not present with the monocytopenia typical of HCL. B symptoms are generally not reported [7,74].

This neoplasm has similar cytomorphologic and immunophenotypic features to those of SMZL when seen in the bone marrow and peripheral blood. However, in the spleen, in contrast to other small B-cell lymphomas, this neoplasm diffusely involves the red pulp, as may be seen in HCL and HCL-v [2,4]. SDRPL patients present with stage IV disease but only rarely show involvement of peripheral lymph nodes [7].

In the peripheral blood, these cells are small in size with mature condensed chromatin, reduced nuclear to cytoplasmic ratio, and mildly basophilic cytoplasm with small, fine cytoplasmic projections reminiscent of those seen in SMZL [72]. SDRPL does not show the plasmacytic differentiation that is sometimes seen in other small B-cell neoplasms [72]. In the bone marrow, neoplastic B-cells typically involve marrow sinusoids, and in the spleen, diffuse involvement of the red pulp cord and sinus infiltration is typical, leading to massive splenomegaly [7,72,73]. Table 3 compares the morphologic features of SRDPL to the other CD5-CD10-LPDs.

**Table 3.** Key morphologic features of CD5-CD10-LPDs, [7,72].

| Neoplasm | Peripheral Blood | Bone Marrow | Spleen |
|---|---|---|---|
| SRDPL | Mature lymphocytes with villous projections | Intrasinusoidal pattern of involvement. No plasmacytic differentiation | Cord and sinusoidal expansion +/− blood lakes |
| HCL | Lymphocytes (less chromatin clumping than normal lymphocytes) with villous projections | Subtle to extensive interstitial infiltrates of lymphocytes. Often extensive reticulin fibrosis | Red pulp cord expansion; atrophic white pulp; blood lakes common. |
| HCL-v | Small lymphocytes with visible nucleoli and villous projections | Intrasinusoidal pattern of involvement; atrophic white pulp | Fill dilated sinusoids +/− blood lakes; atrophic white pulp |
| LPL/WM | Spectrum of lymphocytes, plasmacytoid lymphocytes, and true plasma cells. | Diffuse, interstitial, and nodular involvement by a spectrum of lymphocytes, plasmacytoid lymphocytes, and true plasma cells | Red pulp cord and sinus expansion; atrophic white pulp. |

Abbreviations: HCL-v = hairy cell leukemia variant, LPL/WM = lymphoplasmacytic lymphoma/Waldenström macroglobulinemia, SRDPL = splenic diffuse red pulp small B- cell lymphoma.

Immunophenotypically, SRDPL demonstrates positivity for B-cell markers such as CD20, CD79a, and PAX5. It also expresses DBA44 and IgG but does not characteristically express CD5, CD23, CD10, BCL6, cyclin D1, annexin A1, CD103, CD123, CD11c, CD25, TRAP, or CD43 [3,8]. Cyclin D3 was found in 78% of cases of SDRPL, but in only 2.6% of other small B-cell lymphomas studied [75]. Table 4 differentiates between the immunostains

positive in SDRPL compared to other CD5-CD10-LPDs. Importantly, a complete panel of immunostains is required to differentiate these neoplasms.

**Table 4.** Immunostains in SDRPL compared to HCL, HCL-V, and LPL/WM.

| Antibody | Immunoreactivity in SDRPL | Immunoreactivity in HCL | Immunoreactivity in HCL-V | Immunoreactivity in LPL/WM | Staining Pattern |
|---|---|---|---|---|---|
| CD20 | Yes | Yes | Yes | Yes | Cell membrane |
| CD138 | No | No | No | + in plasma cells | Cell membrane |
| DBA44 | Yes | Yes | Yes | No | Cell membrane |
| IgG | Yes | | Yes | −/+ | Cell membrane |
| Cyclin D3 | Yes | No | No | No | Nuclear |
| TRAP | No | Yes | No | No | Granular cytoplasmic |
| CD11c | −/+ | Yes | Yes | No | Cell membrane |
| CD5 | No | No | No | No | Cell membrane |
| CD123 | No | Yes | +/− | No | Cell membrane |
| Annexin A1 | No | Yes | No | No | Cell membrane |
| CD103 | No | Yes | Variable | No | Cell membrane |
| CD25 | No | Yes | No | +/- | Cell membrane |
| BRAF | No | Yes | No | No | Nuclear |

The molecular findings in SRDPL are under investigation, and recent studies have identified *CCND3* mutations in 26% of patients with SDRPL, but this mutation is found less commonly in other small B-cell lymphomas [5]. *BRAF* mutations, commonly found in HCL, are not typically detected [2,5]. Of note, while this neoplasm typically has a benign course, the presence of *NOTCH1*, *TP53*, and *MAP2K1* mutations have been associated with a more aggressive course [82]. Table 5 compares genetic alterations of *CCND3*, *BRAF*, and *MYD88* in SDRPL, HCL, HCL-V, and LPL.

**Table 5.** Comparison of *CCND3*, *BRAF*, and *MYD88* alterations.

| Abnormality | SDRPL | HCL | HCL-V | LPL |
|---|---|---|---|---|
| *CCND3* mutation | 26% | Negative | 13% | Negative |
| *BRAF* mutation | Negative | >95% | Negative | Negative |
| *MYD88* mutation | Negative | Negative | Negative | >90% |

SDRPL is a malignancy of patients in the seventh decade and is more common in men. SDRPL has a good long-term survival rate of greater than 90%, and more than 85% of patients in one series responded to splenectomy [72,82]. However, this entity does not respond to typical therapies for HCL, HCL-v, and SMZL [73]. Therefore, identification of this entity is required to appropriately direct therapy.

**Funding:** This research received no external funding.

**Acknowledgments:** Special thanks to Ellen F. Lazarus for her support in creating this manuscript.

**Conflicts of Interest:** Disclaimer: The views expressed in this article are those of the authors and do not reflect the official policy of the Department of Army/Navy/Air Force, Department of Defense, or U.S. Government.

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
