# Peer review of "CD5-Negative, CD10-Negative Low-Grade B-Cell Lymphoproliferative Disorders of the Spleen"

_curroncol, doi:10.3390/curroncol28060430_

Round 1
Reviewer 1 Report
Clear and comprehensive review of splenic lymphomas. What I would suggest is that the Authors should reduce the old well known information, focusing on the new less known data (different aspects of splenic lymphomas), possibly extending the sections regarding new therapies
Author Response
Reviewer 1.
Clear and comprehensive review of splenic lymphomas. What I would suggest is that the Authors should reduce the old well known information, focusing on the new less known data (different aspects of splenic lymphomas), possibly extending the sections regarding new therapies.
Thank you for your comments on this manuscript on B cell lymphomas in the spleen. We have added some new information, particularly focusing on lymphoplasmacytic lymphoma as well as treatment for splenic marginal zone lymphoma. However, we are pathologists and our forte is on diagnosis more so than on the details of the newest therapies. Thank you again for your insight.
Reviewer 2 Report
This is a very interesting review of a group of indolent lymphoproliferative disease. Nothing to add.
Author Response
Thank you for your comments
Reviewer 3 Report
This manuscript provided a comprehensive review article regarding the CD5-negative CD10-negative low-grade B-cell lymphoproliferative disorders in the spleen. It mainly focused on SMZL and HCL, and less on HCL-v, LPL, and SDRPL. The manuscript is well-written and easy to understand.
Major comments:
- LPL is a very important differential diagnosis and well-studied. More discussion on this entity is encouraged since the review article has comprehensively discussed SMZL and HCL (also HCL-v) and it would be helpful to include LPL as well. Therefore the reader can have a much deeper understanding about all CD5-CD10-LPD in one good review article.
Minor comments:
- Please provide the references for the statement of incidence of the subtypes of MZL (lines 100-102).
- This sentence “target appearance with proliferation in inner reactive germinal centers as well as outermost mantle zones in cells in marginal zones” (Table 2) needs to be revised.
- Please provide the evidence that HCL-v is an indolent BCL. For my opinion, it is not.
- Please change the images for Figure 4. The blood smear and bone marrow aspirate looked so similar regarding the background cells, though the cytomorphology of the neoplastic cells looked great.
- The manuscript needs careful proofreading, such as CD5-10-LPD in the abstract (line 17, not consistent with CD5-CD10-LPD noted elsewhere), BCL1 or cyclin D1 (better use cyclin D1 consistently), B cells or B-cells, missing all Greek letters missing, typo “patter” in figures 2 and 5 (lines 465 and 625), spaces needed between words (lines 505 and 637), upper vs lower case (lines 626 and 688), gene name in italics (line 255, lines 678-679), and so on.
- The quality of the figures needs to be improved (size too small?), especially the histological pictures in each figure.
Author Response
Reviewer 3.
- LPL is a very important differential diagnosis and well-studied. More discussion on this entity is encouraged since the review article has comprehensively discussed SMZL and HCL (also HCL-v) and it would be helpful to include LPL as well. Therefore the reader can have a much deeper understanding about all CD5-CD10-LPD in one good review article.
Thank you for your comments. I agree that LPL needed to be more fleshed out in the manuscript. This sections has been significantly expanded in the text including defining Waldenstrom macroglobulinemia and discussing some of the prior classifications of this entity.
Minor comments:
- Please provide the references for the statement of incidence of the subtypes of MZL (lines 100-102).
Unfortunately, I was not able to locate the article that I used to quote the different percentages of the subtypes of MZL. Inadvertently, I must have removed the reference during the writing of the manuscript. I agree with you that these percentages should be referenced so I have the language about the percentages from the manuscript since I was unable to find the reference.
- This sentence “target appearance with proliferation in inner reactive germinal centers as well as outermost mantle zones in cells in marginal zones” (Table 2) needs to be revised.
I agree that the wording of this sentence is confusing and could be improved. I have changed the wording to “Positive staining in both the germinal center and marginal zone which has been described as the so-called targetoid pattern.”
- Please provide the evidence that HCL-v is an indolent BCL. For my opinion, it is not.
Thank you for this observation. I removed the word ‘indolent.’ Some pathology literature uses this term in regards to HVL-v, but I do not think it is important in this paper so it has been removed.
- Please change the images for Figure 4. The blood smear and bone marrow aspirate looked so similar regarding the background cells, though the cytomorphology of the neoplastic cells looked great.
The bone marrow aspirate smear is hemodilute, and so the background cells of the image of bone marrow look similar to blood. Note, there is a myelocyte in the bone marrow figure and myelocytes are a normal population in marrow and usually not seen in blood, unless there is some left shift. If you do not like the second image of bone marrow, we can just remove it from the manuscript.
- The manuscript needs careful proofreading, such as CD5-10-LPD in the abstract (line 17, not consistent with CD5-CD10-LPD noted elsewhere), BCL1 or cyclin D1 (better use cyclin D1 consistently), B cells or B-cells, missing all Greek letters missing, typo “patter” in figures 2 and 5 (lines 465 and 625), spaces needed between words (lines 505 and 637), upper vs lower case (lines 626 and 688), gene name in italics (line 255, lines 678-679), and so on.
Yes, some of the authors wrote their sections differently and the paper needs some proofreading. I have proof read the paper to make the adjustments that you mentioned. CD5-CD10-LPD is the abbreviation. Cyclin D1 is used instead of BCL1. B-cell is used instead of B cell.
- The quality of the figures needs to be improved (size too small?), especially the histological pictures in each figure.
Agreed. We have increased the size of the smaller figures. We figure that the editor/Current Oncology staff would adjust the photos to their favored size before publishing.
Thank you again for your comments since they have improved the quality of the paper. All comments have been corrected.